# Intention and Attitude to Accept a Pertussis Cocooning Vaccination among Chinese Children’s Guardians: A Cross-Sectional Survey

**DOI:** 10.3390/ijerph192316282

**Published:** 2022-12-05

**Authors:** Meng Wang, Mengying Li, Xinghui Li, Xiaoli Chen, Feng Jiang, Kezhong A, Zhiguo Wang, Liping Zhang, Yihan Lu, Wenjia Peng, Weibing Wang, Chaowei Fu, Ying Wang

**Affiliations:** 1School of Public Health, Key Laboratory of Public Health Safety, NHC Key Laboratory of Health Technology Assessment, Fudan University, Shanghai 200032, China; 2Shanghai Children’s Medical Center, School of Medicine, Shanghai Jiao Tong University, Shanghai 200127, China; 3Institute of Expanded Programme on Immunization, Guizhou Provincial Center for Disease Control and Prevention, Guiyang 550004, China; 4Institute of Immunization, Qinghai Provincial Center for Disease Control and Prevention, Xining 810007, China; 5Department of Expanded Programmed on Immunization, Jiangsu Provincial Centre of Disease Control and Prevention, Nanjing 210009, China; 6Minhang Center for Disease Control and Prevention, Shanghai 201101, China; 7Department of Epidemiology, Ministry of Education Key Laboratory of Public Health Safety, School of Public Health, Fudan University, Shanghai 200032, China

**Keywords:** pertussis, vaccination, intention, pertussis cocooning vaccination

## Abstract

Objective: to assess Chinese children’s guardians’ intentions and attitudes toward accepting a pertussis cocooning vaccination and its determinants. Methods: a self-administered questionnaire was designed based on a theoretical framework that originated mainly from the reasoned action approach. Associations between questionnaire variables and outcomes were assessed using univariate and multivariate analyses with odds ratios (OR), regression coefficients (β), and their 95% confidence intervals (CIs). Results: among 762 eligible participants, most (80.71%) reported a positive intention to accept a pertussis cocooning vaccination. The guardians’ positive intention was related to the children’s pertussis vaccination experience (OR = 2.483, 95% CI: 1.340–4.600). Guardians who had a positive attitude towards pertussis vaccination (OR = 1.554, 95% CI: 1.053–2.296), higher subjective norms (OR = 1.960, 95% CI: 1.371–2.802) and better perceived behavioral control (OR = 7.482, 95% CI: 4.829–11.591) stated a higher intention to receive a pertussis cocooning vaccination. The mean attitude score was 3.88 ± 0.863. Greater risk perception about pertussis (β = 0.390, 95% CI: 0.298–0.483), stronger obligation from moral norms (β = 0.355, 95% CI: 0.279–0.430), and good knowledge (β = 0.108, 95% CI: 0.070–0.146) were significantly related to positive attitude toward pertussis cocooning vaccination among guardians. Conclusions: Chinese children’s guardians held positive intentions and attitudes toward accepting a pertussis cocooning vaccination. The current findings described the determinants of such intention and attitude and provided knowledge based on improving guardians’ intentions for policymakers if cocooning vaccinations or related immunization strategies are implemented in China in the future.

## 1. Introduction

Pertussis, or whooping cough, is an endemic disease worldwide with a respiratory infection caused by the bacteria *Bordetella (B.) pertussis*. The World Health Organization (WHO) estimates that there were approximately 150,000 cases worldwide in 2018 [1]. From 2004 to 2019, 104,837 cases of pertussis were reported in mainland China, with an increasing incidence over time [2]. According to the monitoring data in 2017, among 194 countries standing the fourth rank in pertussis infections is China [3]. With an epidemic cycle of every 2 to 5 years (typically 3 to 4 years), pertussis still causes severe illness and death among neonates and infants, because they are too young to have completed the primary vaccination series, and pertussis can lead to complications such as pneumonia and encephalopathy and may be disabling or even fatal [4]. Adolescents and adults are the primary sources of *B. pertussis* transmission to unvaccinated infants. A systematic review of the source of infection in infants aged <6 months demonstrated that household contacts were the source of *B. pertussis* in 74–96% of cases where the source was identified. Pooled analysis indicated that, of the household sources, 39% were mothers, 16% were fathers, and 5% were grandparents [5].

Neonatal vaccination has not been adopted because it is controversial whether immunization at birth negatively affects the efficacy of future immunizations [6]. The Global Pertussis Initiative first proposed the strategy of vaccinating adults in close contact with newborns (new mothers, fathers, grandparents, siblings, caretakers, etc.) too young to be fully immunized against pertussis (the “cocooning strategy”) in 2001 [7]. The cocooning strategy was recommended in the United States in 2006, followed by Germany, France, Italy, Costa Rica, and other countries [8]. One study showed that the cocooning strategy averted cases with a higher cost [9]. However, some previous studies have reported that the cocooning strategy did not reduce pertussis in infants [10,11]. Unlike these countries, China does not implement a “cocooning strategy” for immunization and it does not have separate pertussis vaccines for adolescents and adults; this may be one of the reasons why the incidence of pertussis in China has increased in recent years [2].

To date, previous studies have focused on the reasons why the pertussis cocooning strategy is difficult to implement. One study indicated that the cocooning strategy was difficult to implement because of the lack of a platform for postpartum administration of vaccines and programmatic issues in administering vaccines to fathers and other adult household contacts [12]. Another study found that over one-third of respondents indicated that they were mostly concerned about potential side effects of the vaccine on themselves or their infants through breastfeeding or ineffectiveness of the vaccine [13]. In addition, some studies have shown that cocooning strategy had no effect on infant disease [14]. Timely parental pertussis boosters provide significant protection [8]. Guardians’ possible barriers and facilitators of acceptance are crucial. Therefore, the aim of this study was to assess Chinese children’s guardians’ intentions and attitudes toward accepting a pertussis cocooning vaccination and their determinants, to provide specific suggestions to improve the feasibility of the cocoon strategy.

## 2. Method

### 2.1. Study Design and Population

We conducted a cross-sectional study among the guardians of children aged 0–6 years. This specific group was the target population for a possible future pertussis cocooning vaccination program. To be included in the study, guardians had to be at least 18 years of age, a parent or grandparent of a child aged 0–6 years, and had lived in the sample area for six months or more. Those who could not read or write or were unwilling to participate in the study were excluded.

### 2.2. Questionnaire Design

The 63-item questionnaire was developed using a theoretical framework (Figure 1), which included the various determinants of intention to accept a pertussis cocooning vaccination, originating from the results of a literature review and the Reasoned Action Approach (RAA) [15,16].

### 2.3. Variables and Measurements

The primary outcome measure of the questionnaire was the intention of the guardians to accept the cocooning vaccination for pertussis. Participants were asked whether they willing to receive a cocooning vaccination if offered. We measured the personal and psychosocial determinants that potentially influence intention. The personal determinants of intention included personal characteristics (age, sex, education, income, etc.), vaccination experience (guardians’ and children’s pertussis vaccination experience), and pertussis experience (guardians’ and children’s pertussis infection experience). The psychosocial determinants of intention included attitude (a settled way of thinking or feeling about the risk of pertussis infection and the safety and effectiveness of vaccination), subjective norm (which refers to the belief that an important person or group of people will approve and support accepting a pertussis cocooning vaccination), and perceived control (i.e., a person who believes they have the ability and resources to get vaccinated against pertussis).

Attitude was the second main outcome variable, and its determinants included risk perception (people’s subjective judgments about the likelihood of negative occurrences such as pertussis), general vaccination beliefs (reflecting critical vaccination beliefs and including consideration, naturalistic beliefs, and trust in government and industry), outcome expectation (a subjective estimate of how likely it is that receiving a pertussis vaccine will be followed by a particular consequence), and moral norms (a feeling of obligation to adopt a given behavior). Based on the knowledge-attitude-practice model, we added knowledge (about pertussis and its vaccine, such as the source of infection and susceptible people) to the potential determinants of attitude in the theoretical framework for the study. 

We used a five-point Likert scale to measure intention, the psychosocial determinants of intention, and the determinants of attitude (1 = *disagree*, 2 = *somewhat disagree*, 3 = *unsure*, 4 = *somewhat agree*, and 5 = *agree*) [16]. Since intention showed a non-normal distribution (skewness = 1.582, standard error [SE] = 0.089, kurtosis 1.861, SE = 0.177), the intention measure was dichotomized (*somewhat agree* and *agree* were classified as a positive intention; and *unsure*, *somewhat disagree*, and *do not agree* were classified as a unsure/negative intention). Scores for each item were averaged to obtain each of the psychosocial determinants of intention as well as the determinants of attitude-independent categories. Knowledge of pertussis cocooning vaccination was measured using six items. A correct answer was coded as one, whereas incorrect answers were coded as zero. Detailed questionnaire structure and item information are presented in Appendix A. Cronbach’s alpha (α) was 0.869 in this study, indicating acceptable internal consistency.

### 2.4. Data Collection

From 21 July to 22 September 2020, vaccination staff at primary vaccination facilities in the sample districts and counties sent online links to the questionnaire to the guardians of children who came for vaccination (meeting the inclusion criteria). The guardians themselves filled out the brief online questionnaire. According to geographical location, Jiangsu, Guizhou, and Qinghai were selected as sample provinces from eastern, southwest, and northwest China (Figure 2). For each sample province, the provincial capital city with better economic conditions and another city with worse economic conditions were selected as the research objects. The cities of Nanjing (provincial capital city) and Yangzhou in Jiangsu Province, Guiyang (provincial capital city) and Zunyi in Guizhou province, and Xining (provincial capital city) and Haibei in Qinghai province were selected. Three community health service centers and three township health centers were randomly selected from the urban and rural areas of each county, and 15 participants were randomly enrolled from each health center. A total of 762 questionnaires were collected, with a response rate of 99.35%.

### 2.5. Data Analysis

Descriptive statistics were performed by computing summary statistics such as frequencies, percentages, medians, and quartiles. Group comparisons across positive and negative intention participants were conducted using χ^2^-tests and the Mann-Whitney U test. Hierarchical logistic regression analysis was used to investigate the determinants of the pertussis cocooning vaccination. The intention to accept pertussis cocooning vaccination was used as the dependent variable. Independent variables were selected based on the univariate analysis results and previous studies [16,17,18]. We entered the variables in three models: personal characteristics (i.e., guardians’ age, education, household income, and region); vaccination experience (i.e., vaccinated as an adult, and vaccination of own children); and psychosocial determinants of intention (i.e., attitude, subjective norm, and perceived control). Multicollinearity was not observed in the regression analysis (variance inflation factors [VIFs] ranged between 1.06 and 1.08), according to the threshold of <10 suggested by O’ Brien [19].

Univariate and multivariate linear regression analyses were used to examine the determinants of attitudes. Attitude dimension scores were used as dependent variables and independent variables were selected based on previous studies and univariate results [16,20]. Multicollinearity was not observed in the multivariate linear regression analysis (VIFs ranged from 1.03 and 1.63).

All analyses were conducted using STATA version 15.0 (Stata Corporation, College Station, TX, USA), and all figures were analyzed using GraphPad Prism software version 9.0 (GraphPad Software, La Jolla, CA, USA).

### 2.6. Ethics

Ethical approval for this study was granted by Medical Research Ethics Committee, School of Public Health, Fudan University (no. IRB00002408).

## 3. Results

### 3.1. Participants Characteristics

Among the 767 participants, 762 (99.35%) were eligible and included in the analysis. The parents were the main guardians, accounting for 92.91% of the total. Most guardians (67.8%) were women, and nearly half (49.3%) were aged between 30 and 39 years (Table 1). More than half of the guardians had a low education (60%) and low household income (54.3%). Guardians from Jiangsu accounted for the highest proportion (42.8%). The proportion of guardians who had experienced pertussis (1.4%) was similar to that of the children (1.7%). However, the proportion of guardians who had been vaccinated against pertussis (10.9%) was significantly lower than that of the children (56.6%).

### 3.2. Intention to Accept a Pertussis Vaccination

#### 3.2.1. Univariate Analyses

Among the participating guardians, 80.7% reported having a positive intention to accept pertussis vaccination (median = 5, quartile = 4–5). The results of the univariate analyses of personal determinants and intention to accept a pertussis cocooning vaccination are reported in Table 1. The significant determinants of intention were age group, household income, region, pertussis experience in children, guardians’ adult vaccination experience, and children’s pertussis vaccination experience.

The results of the Mann-Whitney U test between psychosocial determinants and intention to accept a pertussis cocooning vaccination are reported in Figure 3. The medians of attitude, subjective norm, and perceived control in the unsure/negative intention group were used as references. The scores of these variables in the positive intention group (median = 4.25, quartile = 3.50–4.75; median = 4.20, quartile = 3.60–4.60; median = 5.00, quartile = 4.00–5.00, respectively) were significantly higher (*p* < 0.001) than those in the unsure/negative intention group (median = 3.00, quartile = 2.75–3.50; median = 3.00, quartile = 2.80–3.40; median = 3.00, quartile = 3.00–3.00, respectively).

#### 3.2.2. Hierarchical Logistic Regression Analysis

The first model, which included personal characteristics (Figure 4; Model I), explained 8.1% of the variance in intention to accept a pertussis cocooning vaccination (R^2^ = 0.081). According to this model, associated with positive intention to accept a pertussis cocooning vaccination were guardians who were aged ≥40 years (OR = 3.601, 95% CI: 1.704–7.610), had a high household income (OR = 1.731, 95% CI: 1.163–2.578), and lived in Guizhou (OR = 2.332, 95% CI: 1.506–3.611) or Qinghai (OR = 2.833, 95% CI: 1.680–4.778).

The second model, which included personal characteristics as well as vaccination experience (Figure 4; Model II), explained 14.7% of the variance in intention to accept a pertussis cocooning vaccination (R^2^ = 0.147). The vaccination experience model added 6.6% to the explained variance, in addition to 8.1% explained by personal characteristics. According to this model, personal characteristics associated with positive intention were as follows: aged ≥40 years (OR = 2.451, 95% CI: 1.134–5.296), high family income (OR = 1.709, 95% CI: 1.130–2.585), and living in Guizhou (OR = 2.084, 95% CI: 1.330–3.267) or Qinghai (OR = 2.669, 95% CI: 1.566–4.548). In addition, guardians who were not clear whether they had received the adult pertussis vaccine were more willing to accept the pertussis cocooning vaccination (OR = 0.473, 95% CI: 0.275–0.812). Children’s experience with the pertussis vaccine was related to their guardians’ positive intentions (OR = 2.643, 95% CI: 1.711–4.082).

The third model incorporated psychosocial determinants from the RAA theory and explained 61.2% of the variance in intention to accept a pertussis cocooning vaccination (R^2^ = 0.612) (Figure 4; Model III). Psychosocial determinants accounted for 46.5% of the explained variance in the intention to accept a pertussis cocooning vaccination based on Model II. Only one vaccine experience variable, that the guardian’s children had been vaccinated for pertussis, remained significant (OR = 2.483, 95% CI: 1.340–4.600). Three psychosocial determinants, attitude (OR = 1.554, 95% CI: 1.053–2.296), subjective norm (OR = 1.960, 95% CI: 1.371–2.802) and perceived control (OR = 7.482, 95% CI: 4.829–11.591) were significantly related to intention.

### 3.3. Attitude to Accept a Pertussis Vaccination

The univariate linear regression analysis showed that all beliefs had a significant influence on attitude (Figure 5). The final multiple linear regression model showed unique contributions to the explanation of attitude toward risk perception (β = 0.390, 95% CI: 0.298–0.483), moral norms (β = 0.355, 95% CI: 0.279–0.430), and knowledge (β = 0.108, 95% CI: 0.070–0.146). The explained variance (R^2^) of the final multivariate model was 35%.

## 4. Discussion

To the best of our knowledge, this study is the first to examine Chinese children’s guardians’ intentions toward cocooning vaccination. Most guardians (80.71%) indicated that they would receive pertussis cocooning vaccination if offered to them. The intention to accept cocooning vaccination influenced children’s vaccination experience, attitude, subjective norms, and perceived control. Further analyses showed unique contributions of guardians’ risk perception, moral norms, and knowledge of pertussis cocooning vaccination in the explanation of attitude.

### 4.1. Intention

The proportion of guardians intending to receive pertussis cocoon vaccination was comparable to that in other studies, with 92.91% being parents. For example, most parents (78%) were willing to accept a pertussis cocooning vaccination if offered to them in a survey among Dutch parents [16]. In countries where pertussis cocooning vaccination is advised, the actual acceptance rate is not high [21,22]. A survey conducted in Germany between 2012 and 2013 showed that 22% of the people living with a baby in one household were vaccinated against pertussis [23]. Clearly, there is a gap between intention and behavior. The data showed that intention predicts only 30–40% of changes in health behavior [24]. Some studies have suggested that the intention-behavior gap originates from variables such as motivation, self-regulation, and habituation [25]. Therefore, these factors should be changed in the future to promote the conversion of intention into behavior.

### 4.2. Determinants of Intention

Surprisingly, we found that one personal determinant influences intention. This finding does not agree with those of previous studies about the intention of cocooning vaccination [16]. According to the present study, guardians whose children had been vaccinated against pertussis were more likely to accept the cocooning vaccination. We hypothesized that guardians whose children had received pertussis vaccination would be more inclined to protect their children from pertussis than guardians whose children had not received pertussis vaccination and would therefore have more positive intentions if offered a pertussis cocooning vaccination.

The theoretical framework of this study included personal determinants, vaccination experience, and the RAA model. This unified model explained 61.2% of the variance in the intention to accept the pertussis cocooning vaccination. Research on the theory of planned behavior has typically treated attitude, subjective norms, and perceived control as independent predictors of intention [26]. A more favorable attitude toward the vaccine could contribute to vaccination acceptance. According to a literature review, attitudes are predictors of the intention to accept vaccination on both the demander and service provider [27]. This result is consistent with our findings. Consequently, the attitude of guardians is an area of concern because if they are not confident enough regarding the cocooning vaccination, this will directly affect vaccine acceptance.

Subjective norm (i.e., the perception that close others approve of vaccination) was the predominant factor in guardians’ intentions. This is consistent with the results of previous studies [28,29]. For example, Wong et al. [13] found that pregnant women were reluctant to get vaccinated owing to the lack of recommendations from healthcare providers. The recommendations of important groups such as doctors, media, and teachers play an important role in the intention of guardians to vaccinate because they have higher education and vaccine-related professional knowledge. The publicity of these groups may improve the intention of guardians to be vaccinated in the future.

Perceived control was strongly related to vaccination intention, consistent with the findings of previous studies [30]. Perceived benefits and barriers were the strongest predictors of behavior. It included whether an individual’s financial status was a barrier to receiving the pertussis vaccine, thus implying that, in the future, the intention of guardians will depend to some extent on the price of the vaccine, which therefore needs to be set within the guardian’s acceptable range.

### 4.3. Determinants of Attitude

Our research confirmed the association of risk perception, moral norms, and knowledge with guardians’ attitudes toward pertussis cocooning vaccination. 

In our study, risk perception was associated with guardians’ attitudes toward the pertussis cocooning vaccination. Previous research has demonstrated that risk perception is an important factor in vaccination intention [31,32] and immunization behavior [33,34]. Immunization research typically includes the perceived susceptibility and severity of vaccine-preventable disease [35], and many studies have shown that perceived vaccination as a cause of adverse effects is consistently associated with vaccine refusal [13]. Therefore, we also included items about side effects in our research. A strong association between low perceived susceptibility and vaccine rejection has been identified; however, evidence for its role in perceived disease severity is weak. This may be because parents first consider whether their child is predisposed to a particular disease before considering its severity [36]. Future research should focus on explaining children’s susceptibility to certain diseases and the side effects of vaccination.

In our findings, moral norms were analyzed as a factor in guardians’ attitudes toward vaccination. Moral norms items were included because the key reason for guardians to be vaccinated against pertussis was to protect their children and to show responsibility. It affects both parental vaccination intention and behavior [37], not only because it predicts both intention and behavior but also serves as a moderator of the intention-behavior relationship [38]. Therefore, future interventions should focus on increasing the strength of moral norms so that guardians feel obligated to be vaccinated to protect their children from pertussis infection. For example, interventions should try to convince guardians that getting vaccinated has an impact not only on themselves, but also on their children, and that vaccination is an efficient way to protect their family from pertussis through health education about the dangers of pertussis disease, as well as the effectiveness of vaccines.

Based on the theory of knowledge-attitude-practice (KAP), knowledge was taken as a potential factor of attitude in the framework for statistical analysis, and the results showed that knowledge significantly affects attitude. This finding is consistent with the results of previous studies [39]. Some studies have reported parental objections to vaccinations for reasons that include inaccurate or inadequate knowledge about the vaccination schedule offered by doctors [40], where to get the vaccine [41], a belief that previous vaccines are still effective, or that one dose is sufficient [42]. Policymakers therefore need to provide effective information on pertussis (pathogens, common signs, and modes of transmission) and guidelines for prevention to help guardians make correct vaccination decisions.

### 4.4. Strengths and Limitations

Based on the population of Chinese children’s guardians, a complete set of determinants was proposed in a robust theoretical framework, and we believe that the explanation framework of intention toward vaccination proposed in this study may help develop pertussis cocooning vaccination programs to optimize vaccination. First, because the framework was carefully and methodically assembled, it was based on a wide range of qualitative and literature, and it conformed to key theoretical concepts about cognitive and emotional factors that determine health behavior, including RAA and KAP. Second, RAA has been the most widely used in vaccine acceptance studies across different disease domains. These theories seem to explain a large proportion of the differences in willingness to accept vaccines. Third, our data show that the constructed theoretical model is robust. Univariate analyses showed that all psychosocial determinants and the determinants of attitude were significantly related to the outcome variable.

However, this study had some limitations. First, some of the questions in our questionnaire may have led to recall bias in respondents’ answers. Second, the exclusion criteria of this study may be a limitation because this group is generally exposed to socio-environmental factors that influence vaccination, as illiteracy may represent low education, financial hardship, lack of health care and other circumstances. Third, there may have been reporting bias in this study. On the one hand, since the questionnaire was answered online and respondents were free to respond; however, since we provided participants with information about pertussis cocooning, the investigators may have been inclined to have positive intentions. Finally, we focused only on the personal and psychosocial determinants of the potential vaccine recipients. However, the environment in which vaccination is provided is vital when implementing a policy strategy. For example, healthcare workers’ attitudes toward pertussis cocooning and financial barriers can also affect actual vaccination. In the future, a wider range of survey participants and factors should be considered.

## 5. Conclusions

We conclude that the intention and attitude toward accepting pertussis cocooning vaccination for guardians in this research is high. Using the robust theoretical framework presented in this study, we identified several determinants that influence vaccination intentions and attitudes. For policymakers, immunization strategies need to be developed by optimizing these determinants in the future to ensure that guardian intention can be translated into action when implementing a pertussis cocooning vaccination strategy. Simultaneously, countries and specific circumstances, such as costs and logistical obstacles, must be considered when developing these programs.

## Figures and Tables

**Figure 1 ijerph-19-16282-f001:**
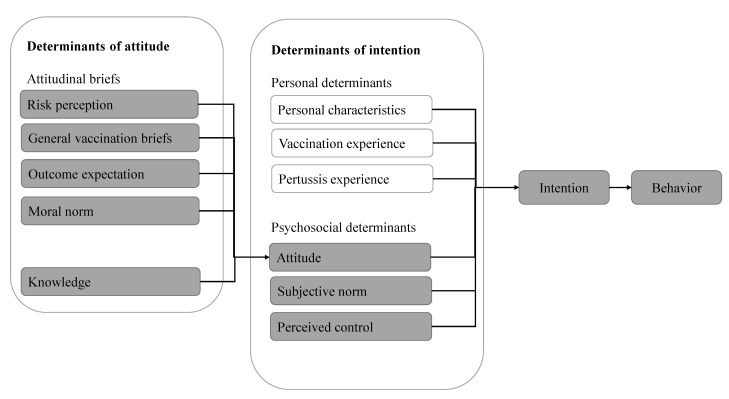
A theoretical framework for accepting a pertussis cocooning vaccination originated from a literature review and the Reasoned Action Approach.

**Figure 2 ijerph-19-16282-f002:**
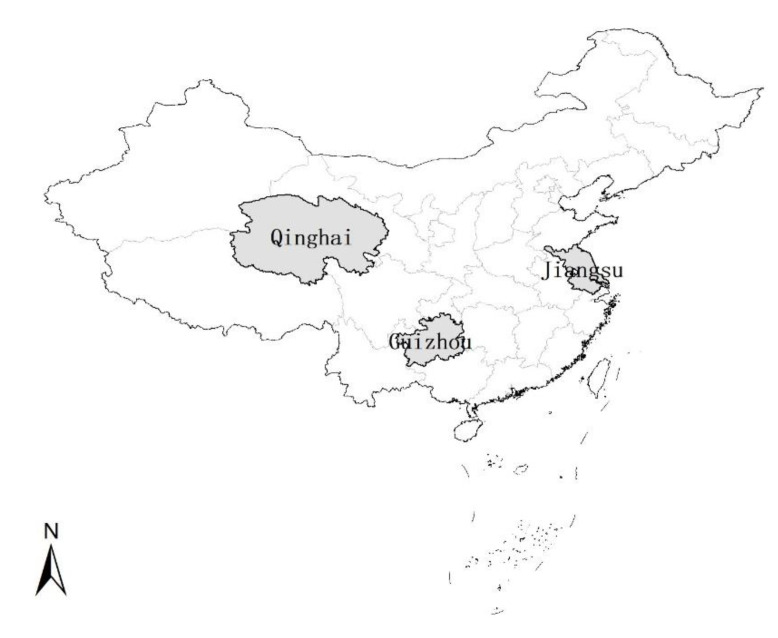
The distribution of the sample provinces in the map of China, and the gray area represents the sample provinces.

**Figure 3 ijerph-19-16282-f003:**
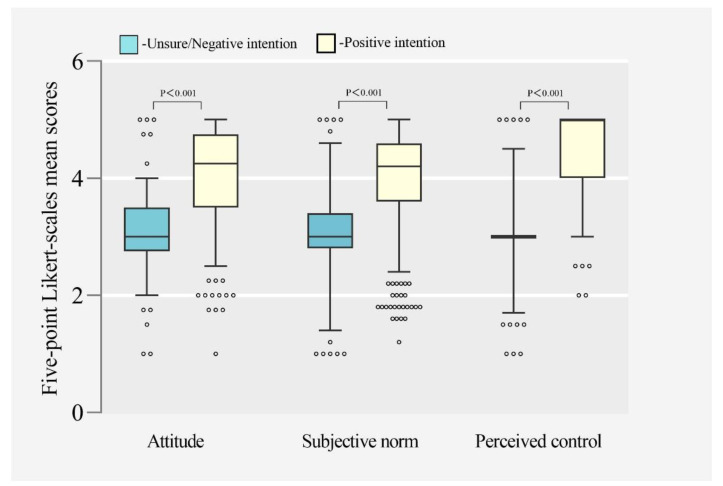
Box plots of five-point Likert scale mean scores of attitudes, subjective norm, and perceived control in the positive intention group and unsure/negative intention group. Data are median (central line), interquartile range (box margins), 5–95% percentile (whiskers), and outliers (dots).

**Figure 4 ijerph-19-16282-f004:**
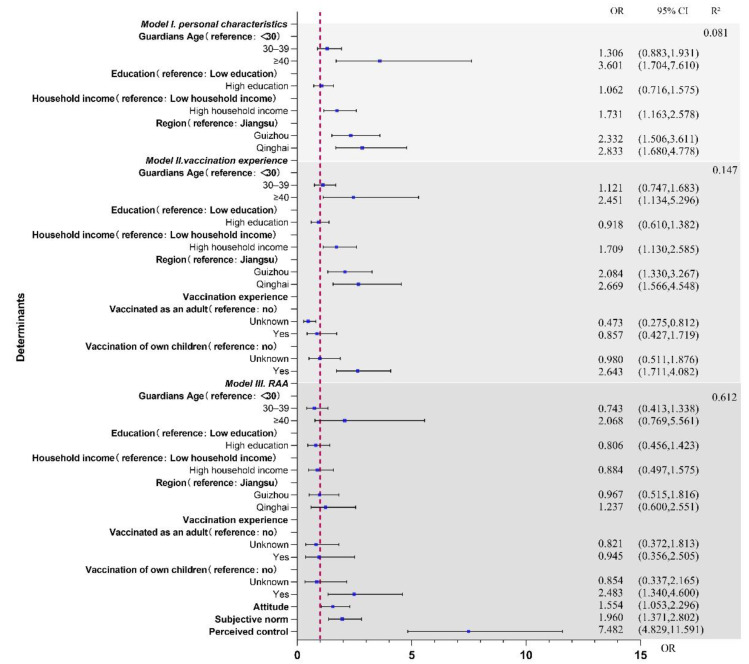
Hierarchical logistic regression analysis of the determinants of intention to accept pertussis cocooning vaccination among guardians. Abbreviations: OR, odds ratio; CI, confidence interval.

**Figure 5 ijerph-19-16282-f005:**
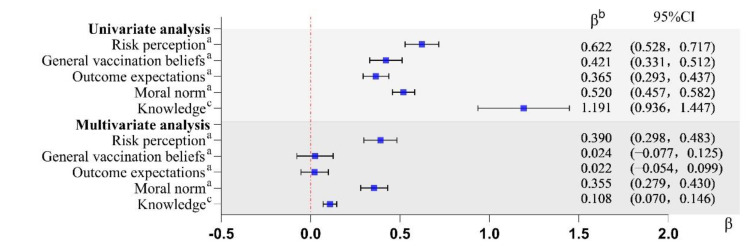
Univariate and multivariate linear regression analysis of the determinants of guardians’ attitude on pertussis cocooning vaccination. Note: CI = confidence interval. ^a^ Measured on a five-point Likert scale; low-high. ^b^ For each point increase on the Likert scale of the determinant, the intention changes with the value of β. ^c^ One point was awarded for each correct answer. R^2^ = 35%.

**Table 1 ijerph-19-16282-t001:** Characteristics of guardians by intention to accept a pertussis cocooning vaccination (N = 762).

Personal Determinants	All Participants (N = 762) n (%)	Unsure/Negative Intention (n = 147, 19.3%) n (%)	Positive Intention (n = 615, 80.7%) n (%)	χ^2^	*p*
**Sex**				**0.539**	0.463
Male	245 (32.2%)	51 (20.8%)	194 (79.2%)		
Female	517 (67.8%)	96 (18.6%)	421 (81.4%)		
**Guardians’ age (years)**				11.236	0.004
<30	284 (37.3%)	68 (23.9%)	216 (76.1%)		
30–39	376 (49.3%)	70 (18.6%)	306 (81.4%)		
≥40	102 (13.4%)0	9 (8.8%)	93 (91.2%)		
**Education**				0.283	0.595
Low ^a^	457 (60%)	91 (19.9%)	366 (80.1%)		
High ^b^	305 (40%)	56 (18.4%)	249 (81.6%)		
**Household income**				4.211	0.04
Low ^c^	414 (54.3%)	91 (22%)	323 (78%)		
High ^d^	348 (45.7%)	56 (16.1%)	292 (83.9%)		
**Region**				17.394	<0.001
Jiangsu	326 (42.8%)	85 (26.1%)	241 (73.9%)		
Guizhou	260 (34.1%)	40 (15.4%)	220 (84.6%)		
Qinghai	176 (23.1%)	22 (12.5%)	154 (87.5%)		
**Pertussis experience**					
Themselves	11 (1.4%)	4 (36.4%)	7 (63.6%)	2.091	0.351
Their children	13 (1.7%)	2 (15.4%)	11 (84.6%)	23.11	<0.001
**Vaccination experience**					
As an adult	83 (10.9%)	12 (14.5%)	71 (85.5%)	12.937	0.002
Their children	431 (56.6%)	50 (11.6%)	381 (88.4%)	40.376	<0.001

*Note:*^a^ junior college or below, ^b^ bachelor’s degree or above, ^c^ ≤5000 CNY monthly household incomes, ^d^ >5000 CNY monthly household income. It refers to net income.

## Data Availability

The data presented in this study are available in article or Appendix A.

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
