# Peer review of "Intention and Attitude to Accept a Pertussis Cocooning Vaccination among Chinese Children’s Guardians: A Cross-Sectional Survey"

_ijerph, 2022, doi:10.3390/ijerph192316282_

Round 1
Reviewer 1 Report
This is a good paper, discussing how effective are mental predispositions in order to bring guardians to vaccinate their children against pertussis. They tried to discover which are the mental inputs that help the most parents or guardians to vaccinate. Thus it is interesting for research and important for public policies.
There is a problem in the title, since it mentions intention and attitude but does not define them or, at least, does not distinguish one from another. Philosophically, these are two concepts very close one to each other. Even though lines 106-127 explain something about each of them, the distinction should be made clearer.
1. This passage in the abstract "Positive 28 attitudes toward pertussis cocooning vaccination (OR = 1.554, 95% CI: 1.053–2.296), higher subjec- 29 tive norm for pertussis vaccination (OR = 1.960, 95% CI: 1.371–2.802), and greater perceived behav- 30 ioral control (OR = 7.482, 95% CI: 4.829–11.591) were significantly associated with positive intention" entails a problem. It is said positive attitudes are "significantly" associated with positive intention. But could it be otherwise? The two parts of the sentence refer to the same attitude (or intention). This should be rephrased.
2. The sentence "In addition, this method is not effective." (line 72) is not clear about which method it refers to. This should be made clear.
3. Is it correct or rather the contrary? "In countries where pertussis cocooning vaccination is advised, the actual acceptance rate is not high" (line 251).
4. "Surprisingly, we found that one personal determinant influences intention. This finding was inconsistent with those of previous studies on the intention of cocooning vaccina-261 tion.[15] Guardians whose children had been vaccinated against pertussis were more likely 262 to accept the cocooning vaccination. We hypothesized that guardians whose children had 263 received pertussis vaccination would be more inclined to protect their children from per-264 tussis than guardians whose children had not received pertussis vaccination and would 265 therefore have more positive intentions if offered a pertussis cocooning vaccination." I would suggest to make this passage clearer; it could be changed to: "Surprisingly, we found that one personal determinant influences intention. This finding questions those of previous studies about the intention of cocooning vaccina-261 tion.[15] According to the present study, guardians whose children had been vaccinated against pertussis were more likely 262 to accept the cocooning vaccination. We hypothesized then that guardians whose children had 263 received pertussis vaccination would be more inclined to protect their children from per-264 tussis than guardians whose children had not received pertussis vaccination and would 265 therefore have more positive intentions if offered a pertussis cocooning vaccination."
I think the paper sometimes lacks a change in its form, so that it will be clear if a sentence corroborates the preceding one or contradicts it. But actually what I would recommend is more common in Romance languages than in English.
Reviewer 2 Report
Dear Author,
It was a pleasure to read your manuscript. It presents very important research findings, both from a societal and health perspective. All my comments can be found in the appendix.

Round 2
Reviewer 2 Report
Dear Authors,
thank you for your conscientious response to my comments. I believe that the manuscript as it stands meets the requirements for publication. I congratulate you on a good and highly relevant study.
Best regards,
Reviewer